# OrganITE: Optimal transplant donor organ offering using an individual treatment effect

**Jeroen Berrevoets**
University of Cambridge,
Vrije Universiteit Brussel
jeroen.berrevoets@maths.cam.ac.uk

**James Jordon**
University of Oxford
james.jordon@wolfson.ox.ac.uk

**Ioana Bica**
University of Oxford,
The Alan Turing Institute
ioana.bica@eng.ox.ac.uk

**Alexander Gimson**
Cambridge University Hospitals
alexander.gimson@nhs.net

**Mihaela van der Schaar**
University of Cambridge,
University of California, Los Angeles,
The Alan Turing Institute
mv472@cam.ac.uk

## Abstract

Transplant-organs are a scarce medical resource. The uniqueness of each organ and the patients' heterogeneous responses to the organs present a unique and challenging machine learning problem. In this problem there are two key challenges: (i) assigning each organ "optimally" to a patient in the queue; (ii) accurately estimating the potential outcomes associated with each patient and each possible organ. In this paper, we introduce OrganITE, an organ-to-patient assignment methodology that assigns organs based not only on its own estimates of the potential outcomes but also on organ scarcity. By modelling and accounting for organ scarcity we significantly increase total life years across the population, compared to the existing greedy approaches that simply optimise life years for the current organ available. Moreover, we propose an individualised treatment effect model capable of addressing the high dimensionality of the organ space. We test our method on real and simulated data, resulting in as much as an additional year of life expectancy as compared to existing organ-to-patient policies.

## 1 Introduction

For patients with endstage diseases, organ-transplantation surgery is often a last resort. With an increasing number of donors and patients in need of a transplant [1, 2]– organ transplantation surgeries have doubled in the last fifteen years [3] –finding the best recipient for each available organ is becoming an ever more important problem. Though the problem is one of treatment effect estimation and resource allocation, unique challenges arise in each sub-problem in the setting of organ transplantation.

**The problem.** There are several possible objectives we could consider optimising in the setting of organ matching, e.g. total life years of the population, number of deaths in the waiting queue, deaths before 5 years after transplant, etc. We focus on the objective of *maximising the total life*

*years* of the population (or equivalently the mean life expectancy), a discussion on the morality of this objective is provided in our broader impact section at the end of this paper. Even with a clearly defined metric, assigning organs to patients involves a challenging balance of several aspects: (i) optimality in this setting is subject to a Pareto efficiency - assigning an organ to a patient implies *not* assigning the organ to other patients; (ii) each organ is *unique* and high-dimensional, thus rendering outcome estimation for each (also unique) patient very difficult; (iii) organs arrive in a stream - while the current organ might result in a positive outcome for a patient, future organs might have an even greater positive outcome (and we do not know which organs will become available in the future); (iv) each patient will die soon if not given an organ and thus has (potential) access to only a limited number of organs.

**Transplant surgery is an invasive procedure.** Besides rarity of an organ, an organ-to-patient assignment policy should be based on the *difference* between transplanting the organ and not, rather than either outcome alone. Both outcomes will therefore need to be estimated, though no paired observations are present in the data - all patients either received or did not receive an organ. This is the well-studied problem of individualised treatment effect (ITE) estimation [4–7].

In significant contrast to our setting, though, most existing work for ITE estimation focuses on a single binary treatment (e.g. [4, 5, 7]). Few methods exist that can naturally handle even multiple categorical treatments (e.g. [6]). Methods addressing the continuous setting (e.g. treatments with a dosage) have typically only been applied to one-dimensional treatments (e.g. [8]). Performing ITE estimation in organ transplantation is non-trivial for several reasons: (i) organs - our treatments - are high dimensional, with a mixture of continuous and categorical features; (ii) observational data for organ transplantation is highly biased towards organ-assignment policies already in place; (iii) organs are unique and limited - each patient's ideal organ may not be available to them. It is exactly this last challenge that makes the problem of transplant-organ assignment so interesting.

**Estimating outcomes is not enough.** Contemporary organ-to-patient matching policies assign an organ to the patient based on estimated outcomes (e.g., life expectancy or survival probability) [9]. Indeed, by increasing outcome prediction accuracy, one can better optimise relevant metrics such as life expectancy [10]. However, such policies fail to account for the arrival distribution of organs and thus ignore (iii) and (iv) above. The importance of how rare an organ is, is a direct result of every patient's deteriorating condition. When a particular patient is in need of a rare type of organ, they might need to settle for a less suitable organ that is available *now*. Moreover, the organ may be a better fit for someone else, but that other patient may be in better condition and thus able to wait longer for a second suitable organ to be available. Such a consideration is completely ignored when a policy focuses solely on a patient's potential outcomes. Figure 1a illustrates why organ density is an important consideration and Figure 1b compares existing policies' considerations with OrganITE's. While rarity of an organ has previously been taken into account through sensitization levels in the kidney exchange setting [11], OrganITE is the first to explicitly model the organ density.

**Our contribution: OrganITE.** In this paper we propose a high-dimensional ITE estimator and an organ-to-patient assignment policy that accounts for organ rarity. For our ITE estimator, we create a latent space of patient-organ pairs, balanced using domain adversarial training, upon which we build an outcome predictor. For our organ-to-patient policy, we rank patient-organ pairs based on three criteria: (i) the organ-to-patient match in terms of an organ's ITE on life expectancy, (ii) how distant the organ is to the patient's optimal organ, and (iii) organ rarity. We conduct experiments on a mixture of real and synthetic data, with the real data consisting of 26 years of organ transplantation surgery in the UK (approx. 19k patients). Our experiments demonstrate the efficacy of our method, by demonstrating a higher life expectancy and fewer deaths, both before *and* after transplant surgery, in comparison to other policies.

## 2   Related work

**ITE estimation.** An important component in our organ-to-patient assignment policy, is a high dimensional ITE estimator. In ITE estimation, most research is devoted to coping with assignment bias, i.e., when individuals are granted a treatment, the per-treatment outcome estimate is a biased estimate as treatments are usually not assigned in a random fashion. This is especially true in our

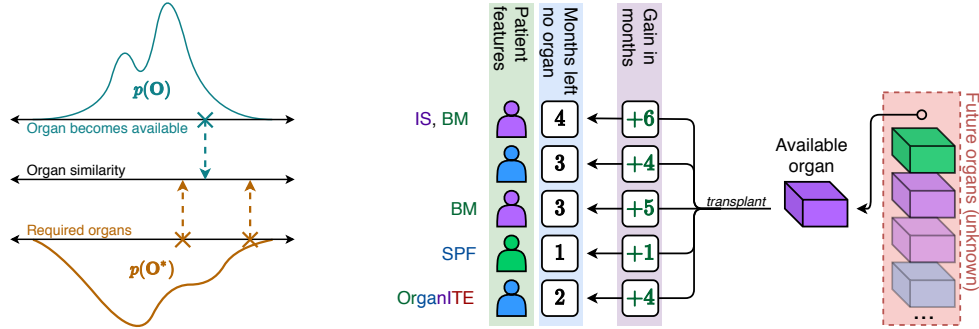

(a) Consider the density of organs, $p(\mathbf{O})$, above, and the density of required organs, $p(\mathbf{O}^*)$, below. While the available organ might be a better fit for the leftmost patient as it is similar to their required organ, we argue the available organ might be best assigned to the rightmost patient as their required organ is less probable to become available.

(b) Above figure illustrates a transplant system, with four policies: Incremental Survival (IS), Best Match (BM), Sickest person first (SPF), and OrganITE. Each policy is colored according to their input from the system, and placed next to the patient they would select from the waiting queue given the available organ. Contrasting other policies, OrganITE takes into account every column by estimating an ITE of the organ on every patient's life expectancy, and balances this ITE with the density of future organs and a patient's ideal transplant-organ.

Figure 1: Method overview: in Figure 1a we illustrate the importance of an organ density in organ-to-patient assignment; in Figure 1b we compare OrganITE with other, known, policies in organ-to-patient assignment.

setting where random organ-to-patient assignment would indeed yield unbiased ITE estimates, but at the price of many lives.

In its simplest form, a binary or categorical ITE estimator predicts outcomes given a treatment indicator [4, 5, 12–15]. Assignment bias then manifests itself as a covariate shift of the individuals associated with different treatment indicators in the data. Handling this bias is done by building a balanced representation based on discrepancy measures between the subsets associated with different treatments [4, 7, 16–18]. Of particular interest to our work is Bica et al. [19], who use domain adversarial training [20–22] of the treatments to learn this representation.

Above methods require the explicit definition of different treatment-based subsets in the population. When the treatment is a continuous variable, e.g., a dosage, it is unclear how to correctly divide the population in distinct subsets. One approach is to use the generalised propensity score [23, 24] or to discretise the treatment dosage in intervals [25]. Bica et al. [8] propose a different method as they generalise the GANITE framework [6] to generate counterfactual values for each individual, given a treatment dosage.

We found TransplantBenefit (TB) [9], the current United Kingdom endstage liver disease assignment policy, to use a naive— high-dimensional —ITE estimator. Neuberger et al. [9] use simple linear models to fit two distinct models: using only patients that died prematurely; and using patients that were granted an organ [26, 27]. Using both models, the patient with the largest difference between both outcomes, receives the organ. We identify two main disadvantages to this approach: (i) in no way is assignment bias accounted for; (ii) linear models have a very limited capacity which require careful feature engineering.

**Organ matching.** Work in machine learning devoted to organ matching is focused on a personalised objective. For example, ConfidentMatch [10, 28] introduced a cluster-based scheme whereby the patients-organ pairs were divided into different clusters, upon which a predictor was trained to estimate the respective outcome of pairing the patient with the given organ. Similarly, Pérez-Ortiz et al. [29] introduced an extension on the model for endstage liver disease (MELD) [27] by using a support vector machine as well as ordinal linear regression as an estimator on patient-organ pairs, rather than patients alone. These methods differ much from our work in two main regards: (i) there is no notion of ITE, as there is no outcome prediction for the patient without an organ [28]; (ii) contrasting our approach, we do not necessarily grant the organ to the most suitable patient, i.e. the one yielding largest survival time, but also consider the probability of necessary organs of other patients.

**Resource allocation.** As we are tasked with assigning a scarce resource, while respecting some Pareto efficiency, we touch on resource allocation [30, 31]. However, resource allocation as a field is incredibly broad and finds applications in Economics and Operations Research [32, 33], Networks [34, 35], and Healthcare [36–38]. While most work in Healthcare is focused towards monetary allocation, some work is concerned with allocating scarce medical treatment to patients, e.g., ICU beds [39], Influenza vaccine in a pandemic crisis [40] or organ transplantations [41]. In fact, the organ matching polices described in Neuberger et al. [9] align very well with the allocation strategies discussed in Persad et al. [42], who categorises allocation strategies in four subgroups: treating people equally, favouring the worst-off, maximising total benefits, and promoting and rewarding social usefulness.

Allocating scarce medical treatment brings forth an ethical discussion. In fact, many of the recent contributions in medical resource allocation have ethics as their subject [43–45]. In this work, however, we propose a method that optimises both death-rates as well as life years gained, corresponding to a utilitarianism point of view for its ethics, as was introduced by Persad et al. [42].

## 3 Problem formulation

**Notation.** Let $\mathcal{X} \subset \mathbb{R}^d$ denote the space of all possible patients and $\mathcal{O} \subset \mathbb{R}^e$ denote the space of all possible organs. Let $\mathbf{X} \in \mathcal{X}$ denote the feature vector of a patient. We adopt the Rubin-Neyman potential outcomes framework [46], defining $Y^{\mathbf{o}}$ to be the (potential) life-expectancy associated with transplanting organ $\mathbf{o} \in \mathcal{O} \cup \{\emptyset\}$ into patient $\mathbf{X}$, where $\mathbf{o} = \emptyset$ indicates the outcome when no organ is given (the patient died before receiving a transplant). Let $\mathbf{O} \in \mathcal{O} \cup \{\emptyset\}$ denote the organ (or no organ) assigned to $\mathbf{X}$, and let $Y = Y^{\mathbf{O}}$ denote the observed outcome. We denote the optimal organ for patient $\mathbf{X}$ as $\mathbf{O}_{\mathbf{X}}^* = \arg\max_{\mathbf{o} \in \mathcal{O}} Y^{\mathbf{o}}$.

For $t \in \mathbb{Z}^+$, let $\mathcal{X}^t \subset \mathcal{X}$ denote the set of patients in the transplant system at time $t$. We divide each $\mathcal{X}^t$ into 2 disjoint sets: $\mathcal{X}_Q^t$ (the waiting queue) and $\mathcal{X}_M^t$ (the monitored set). A patient, $\mathbf{x}$ (sampled from some distribution $p(\mathbf{X})$), enters the system by first joining $\mathcal{X}_Q^s$ at some time $s = s_{\mathbf{x}}$ when it is determined they need a transplant. They remain in the system until either they die, or they receive a transplant (i.e. $\mathbf{x} \in \mathcal{X}_Q^{s+j} \forall j < k$ for some $k$). If they receive a transplant, they move to $\mathcal{X}_M$ (i.e. $\mathbf{x} \in \mathcal{X}_M^{s+k}$). At each time-step, an organ $\mathbf{O}_t \in \mathcal{O}$ becomes available for transplant according to some distribution $p(\mathbf{O})$ and a patient $\mathbf{X}_t \in \mathcal{X}_Q^t$ is selected to receive it (moving the patient to $\mathcal{X}_M^{t+1}$) according to some policy $\pi$. For a patient, $\mathbf{X}$, we denote by $\mathbf{O}_{\mathbf{X}}^{\pi} \in \mathcal{O} \cup \{\emptyset\}$ the organ assigned to that patient under the policy $\pi$.

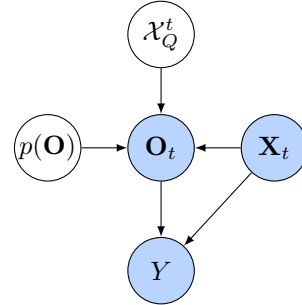

Figure 2: Graphical structure of one patient-organ pair. $\mathbf{X}$ influences $\mathbf{O}_{\mathbf{X}}$ and $Y$. $\mathbf{O}_{\mathbf{X}}$ is also influenced by other patients in the queue, $\mathcal{X}_Q$ and the arrival distribution $p(\mathbf{O})$.

**Training data.** We assume access to a dataset of past organ assignments alongside patient outcomes. Formally, we have access to $\mathcal{D} = \{(\mathbf{x}_i, \mathbf{o}_i, y_i) : i = 1, ..., N\}$, where each tuple $(\mathbf{x}, \mathbf{o}, y)$ is generated according to $p(\mathbf{X}), p(\mathbf{O})$ and $\pi_{obs}$, where $\pi_{obs}$ is the existing policy followed by clinicians. Note that we assume access to each of $p(\mathbf{X}), p(\mathbf{O})$ and $\pi_{obs}$ through the training dataset. For any patient features $\mathbf{x} \in \mathcal{X}$ we define the optimal-in-the-dataset organ by $\hat{\mathbf{O}}_{\mathbf{x}}^* = \arg\max_{\mathbf{o} \in \mathcal{D}} \mathbb{E}(Y^{\mathbf{o}} | \mathbf{X} = \mathbf{x})$.

**Objective.** Our goal, given $\mathcal{D}$, is to produce a policy $\hat{\pi} : \mathcal{P}(\mathcal{X}) \times \mathcal{O} \to \mathcal{X}_Q$ that maximises $\mathbb{E}(Y^{\mathbf{O}_{\mathbf{x}}^{\hat{\pi}}})$ where the expectation is taken over $p(\mathbf{X})$ and $p(\mathbf{O})$.

**Evaluation.** The performance of a policy is reported in total population life years over 26 years of organ transplants. To better understand this number, we provide supplemental material reporting: average time alive– in $\mathcal{X}_Q$ and $\mathcal{X}_M$; and premature deaths, where any death in $\mathcal{X}_Q$, and a death sooner than 5 years in $\mathcal{X}_M$ is considered premature. Note that for average time alive in a set, we consider only patients that died while in this set, i.e., when a patient is moved from $\mathcal{X}_Q$ to $\mathcal{X}_M$, their time spent in $\mathcal{X}_Q$ does not count towards the average time alive of $\mathcal{X}_Q$.

**Assumptions.** The first step in producing a policy from the training data will involve estimating the potential outcomes. Here we state the standard assumptions made to perform ITE estimation.

**Assumption 1 (Overlap.)** *For all* $\mathbf{x} \in \mathcal{X}$ *and all* $\mathbf{o} \in \mathcal{O}$, $0 < p(\mathbf{o}|\mathbf{x}) < 1$.

With this assumption every patient has a non-zero probability of receiving any organ from the organ-space, $\mathcal{O}$.

**Assumption 2 (Unconfoundedness.)** *Conditional on* $\mathbf{X}$, *the full set of potential outcomes are independent of the treatment,* $\mathbf{O}_{\mathbf{X}}^{\pi_{obs}} \perp \!\!\! \perp \{Y^{\mathbf{o}} : \mathbf{o} \in \mathcal{O}\}|\mathbf{X}$, *where* $\mathbf{O} \in \mathcal{O}$ *is a realisation of an organ from the organ-space,* $\mathcal{O}$.

Following these assumptions, we can describe the data using a simple graphical structure as in Figure 2. Given this graphical structure, a large focus in ITE estimation is deconfounding the treatment from a patient in observational data, i.e., to correctly estimate the potential outcome, $\mathbb{E}(Y|do(\mathbf{O}), \mathbf{X})$, we must deconfound the relationship between $\mathbf{O}$ and $\mathbf{X}$ [47]. This is especially important in our case, as there is a strong bias due to present policies deciding who will receive what organ.

## 4  OrganITE

### 4.1  Organ-to-patient assignment

At time $t$, given a new organ $\mathbf{O}_t$, we score every patient in the waiting list, $\mathcal{X}_Q^{t-1}$. Our score balances: **(i) ITE** on life expectancy; **(ii) distance** to $\hat{\mathbf{O}}_{\mathbf{X}}^*$, our approximation of the best organ; and **(iii) rarity** of this best organ. Formally, we select $\mathbf{X}_t$ according to:

$$\mathbf{X}_t = \hat{\pi}(\mathcal{X}_Q^t, \mathbf{O}) := \underset{\mathbf{X} \in \mathcal{X}_Q^t}{\arg\max} \left\{ \lambda(\mathbf{O}, \mathbf{X}) \Big[ \underbrace{\alpha_1 S(\mathcal{X}_Q^t \setminus \{\mathbf{X}\}) + \alpha_2 S(\mathcal{X}_M^t \cup \{\mathbf{X}\})}_{\text{(i) ITE}} \Big] \right\}, \quad (1)$$

where $S(\mathcal{X}') := \sum_{\mathbf{X} \in \mathcal{X}'} \mathbb{E}(Y|\mathbf{X})$ denotes the life expectancy of those in the set $\mathcal{X}'$; $\alpha_1, \alpha_2 \in \mathbb{R}_+$ are hyperparameters; and distance and rarity are incorporated through $\lambda$ defined by:

$$\lambda(\mathbf{O}, \mathbf{X}) := \Big[ \underbrace{\text{Dist}(\mathbf{O}, \hat{\mathbf{O}}_{\mathbf{X}}^*)^{-a}}_{\text{(ii) distance}} \underbrace{p(\hat{\mathbf{O}}_{\mathbf{X}}^*)^{-b}}_{\text{(iii) rarity}} \Big], \quad (2)$$

where $a, b \in \mathbb{R}$ are hyperparameters governing the balance between distance and rarity.

**Hyperparameters.** Selecting good values for the hyperparameters introduced above takes careful thought as they reflect a morality in organ-assignment. For example, higher $\alpha_1$ could lead to less-sick patients potentially getting treatments early, resulting in lower population life-years overall. Likewise for $a$ and $b$, as they impact whether or not the policy should care more about those who require a rare organ, or those who are best fit to receive the available organ. As with all hyperparameters, we report our settings in the supplemental material.

Each of the three components in (1) needs to be estimated from data, as they are unknown. Using standard methods, we first discuss (ii) distance and (iii) rarity. We then discuss (i) ITE in Section 4.2.

**(ii) distance: finding the optimal organ.** Using an ITE estimator (described in the next section) to approximate $Y$, we find $\hat{\mathbf{O}}_{\mathbf{x}}^* = \arg\max_{\mathbf{o} \in \mathcal{D}} \mathbb{E}(\hat{Y}^{\mathbf{o}}|\mathbf{X} = \mathbf{x})$, where $\hat{Y}$ is the estimate of $Y$. In our experiments we define $\text{Dist}(\cdot, \cdot)$ to be the Euclidean distance.

**(iii) rarity: estimating the organ density.** We use two strategies to model the organ density: a standard undercomplete auto-encoder paired with a Gaussian kernel density estimator (KDE), sometimes referred to as a Parzen-Rosenblatt window [48, 49]; and a variational auto-encoder (VAE) [50]. We use these strategies to estimate $p(\mathbf{O}_{\mathbf{X}}^*)$, used in (2). Hyperparameters for these densities follow standard procedures which we report in our supplemental material.

### 4.2  ITE estimation

To compute the ITE term in (1) we are tasked with estimating the effect of transplanting the current organ into each patient in the current queue. While the other components in OrganITE– distance

and rarity —are estimated using standard methods, estimating an ITE is much more involved as we need to account for assignment bias in the uniquely challenging setting of the high dimensional treatment space (the organ space). Without balancing this bias we cannot use our network to predict out-of-sample life expectancy for patient-organ pairs [4, 51].

**Balancing a representation.** Our ITE estimator learns a balanced representation, $\phi$, of the patient-organ pair through domain adversarial training [20] of the propensity score of an organ. To build these propensities, we reduce the dimensionality of organs by introducing organ clusters, $c_i(\mathbf{O}), i = 1, ..., k$. We consider a representation balanced when $\phi$ is invariant across organs, $p(\phi|c_1(\mathbf{O})) = \cdots = p(\phi|c_k(\mathbf{O}))$. To build organ invariant representations while also estimating patient outcomes, we aim to maximize propensity loss and minimize outcome loss. When the representation is invariant, we have effectively removed assignment-bias as the organ-type no longer hints towards any particular patients.

Let $\text{ITE}_{\theta_\Phi,\theta_p,\theta_Y}$ be our ITE estimator, with $\theta_\Phi, \theta_p, \theta_Y$ as the parameters for: the representation, $\phi = \Phi_{\theta_\Phi}(\mathbf{X}, \mathbf{O})$; the propensity, $\hat{c} = c_{\theta_p}(\phi)$; and the outcome, $\hat{Y} = Y_{\theta_Y}(\phi)$. Using these parameters we define: a cross-entropy loss (CE) for the propensity estimator,

$$\mathcal{L}_{\text{CE}}(\theta_\Phi, \theta_p) :=$$
$$-\sum_i^k \mathbb{I}_{\{c(\mathbf{O})=\hat{c}\}} \log \left( c_{\theta_p}(\Phi_{\theta_\Phi}(\mathbf{X}, \mathbf{O})) \right), \quad (3)$$

and a mean squared error loss (MSE) for the outcome estimator,

$$\mathcal{L}_{\text{MSE}}(\theta_\Phi, \theta_Y) := ||\hat{Y} - Y||_2^2. \quad (4)$$

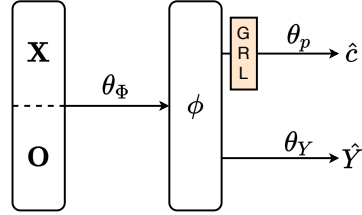

Figure 3: Our ITE network: we concatenate the organ, $\mathbf{O}$, to the patient, $\mathbf{X}$, to learn a representation, $\phi$. We handle assignment bias by estimating the propensity, $p(c_i(\mathbf{O})|\phi)$, after a gradient reversal layer (indicated by GRL in yellow), and estimate the outcome, $Y$. When no organ is provided, and $\mathbf{O} = \mathbf{0}$, the network outputs $Y^\emptyset$.

We then subtract (3) from (4) to arrive at our final loss,

$$\mathcal{L}_{\text{ITE}}(\theta_\Phi, \theta_p, \theta_Y) := \mathcal{L}_{\text{MSE}}(\theta_Y, \theta_\Phi) - \gamma \mathcal{L}_{\text{CE}}(\theta_p, \theta_\Phi), \quad (5)$$

where the hyperparameter $\gamma \in [0, 1]$ controls the trade-off between domain discrimination and outcome prediction.

By optimising the parameters over (5) we reach a saddle point $(\theta_\Phi^*, \theta_p^*, \theta_Y^*)$. Setting the model's parameters to this saddle is shown (in [19]) to result in the balance between domain discrimination and outcome prediction,

$$\theta_p^* := \underset{\theta_p}{\arg\max} \, \mathcal{L}_{\text{ITE}}(\theta_\Phi^*, \theta_p, \theta_Y^*) \quad \text{and} \quad (\theta_\Phi^*, \theta_Y^*) := \underset{\theta_\Phi, \theta_y}{\arg\min} \, \mathcal{L}_{\text{ITE}}(\theta_\Phi, \theta_p^*, \theta_Y). \quad (6)$$

We learn these parameters by using a gradient reversal layer (GRL) [20, 21] before the parameters of the propensity estimator, $\theta_p$, as is illustrated in Figure 3. This allows us to learn the ITE-model using backpropagation to minimise (5). A full training algorithm is provided in our supplemental material.

**Using the model.** Rather than estimating the life expectancy for an organ's cluster, $c_i(\mathbf{O})$, we input the organ, $\mathbf{O}$, directly concatenated to the patient, $\mathbf{X}$– illustrated in Figure 3. To estimate life expectancy without the organ, a zero vector is concatenated instead.[1] As such, the organ-clusters are only used to estimate a propensity score, contrasting Li et al. [22] where they use the clusters to estimate counterfactuals also, albeit in a one-dimensional setting.

**Hyperparameters.** We learn the organ-clusters using a KMeans-cluster with a Euclidean distance. While the amount of clusters will be dependent on the treatment, we found 15 clusters to work well in our setting as it resulted in a low root mean squared error (RMSE) on the outcome predictions. Besides standard hyperparameters for the neural network in Figure 3 such as: layer count, width, and activation functions; our model requires additional attention to $\gamma$ which controls the equilibrium in (6). For example, when $\gamma = 0$, the trade-off is completely skewed towards high prediction on the

dataset, without balancing the representation through, $\theta_p$, resulting in poor out-of-sample prediction due to assignment bias. When $\gamma = 1$, we find the prediction to suffer too much, resulting in poor convergence. Setting this parameter requires domain knowledge on the assignment bias, as the out-of-sample performance is based on (unobservable) counterfactuals. We refer to our supplemental material for our hyperparameter settings.

## 5 Experiments

### 5.1 ITE estimation

**Experimental setup.** We train our model using observational data— with bias in the organ assignments —and evaluate the model on how well it estimates the patient outcomes under different possible organ options. Rather than *making up* a bias over the patient-organ pairs, we simply use real data, but replace the outcome with, $Y^{\mathbf{O}_{\mathbf{X}}^{\pi_{obs}}} = \nu \exp\{\boldsymbol{\theta}_1^T \mathbf{X} + \boldsymbol{\theta}_2^T \mathbf{O}_{\mathbf{X}}^{\pi_{obs}} + \frac{1}{2}\} + \mathcal{N}_Y$ (cfr. Figure 2), where $\nu$ is a scalar, scaling the output of the function such that they share the same average as in the data, and $\mathbf{O}_{\mathbf{X}}^{\pi_{obs}} = \mathbf{0}$ for an outcome without the organ. A similar strategy for the binary treatment setting was used in Alaa and van der Schaar [5]. We report performance on the normalised root mean squared error (NRMSE), where lower is better.

**Results.** In Table 1 we report an ablation study on different network architectures: a padded network, as introduced in Section 4.2; a siamese network where we lead the inputs with, and without, organ to the latent space; and two distinct networks, one for outcome prediction with organ, and one without. We find the padded network to outperform the others and thus use it as our ITE model throughout the remainder of our experiments. Using our padded network, in Table 2 we compare favorably against: ConfidentMatch [10], which we updated for ITE estimation by using two models, as well as a padded version as we did in our ITE estimator; and a multitask network, predicting the outcome for the organ-clusters. More details regarding the benchmarks, as well as the ablation networks can be found in our supplemental material.

Table 1: NRMSE for ITE estimation on synthetic data over 10 different folds, standard deviation is reported in brackets below each sore.

|  | Padded network | | Siamese network | | Two networks | |
|---|---|---|---|---|---|---|
|  | 7 clusters | 15 clusters | 7 clusters | 15 clusters | 7 clusters | 15 clusters |
| $\gamma = 0$ | 0.1926 ($\pm$ 0.01) | **0.1845** ($\pm$ 0.01) | 0.3651 ($\pm$ 0.02) | 0.3480 ($\pm$ 0.01) | 0.3913 ($\pm$ 0.02) | 0.3463 ($\pm$ 0.03) |
| $\gamma = 0.15$ | 0.1837 ($\pm$ 0.02) | **0.1550** ($\pm$ 0.02) | 0.2094 ($\pm$ 0.03) | 0.2345 ($\pm$ 0.02) | 0.3822 ($\pm$ 0.04) | 0.2730 ($\pm$ 0.05) |

Table 2: NRMSE for ITE estimation on synthetic data over 10 different folds, standard deviation is reported in brackets below each sore.

|  | ConfidentMatch | | Multitask network for $c(\mathbf{O})$ | OrganITE($\gamma = 0.15$) |
|---|---|---|---|---|
|  | Two models | Padded |  |  |
| 7 clusters | 0.5678 ($\pm$ 0.02) | 0.2356 ($\pm$ 0.02) | 0.5619 ($\pm$ 0.03) | **0.1837** ($\pm$ 0.02) |
| 15 clusters | 0.6372 ($\pm$ 0.03) | 0.5675 ($\pm$ 0.05) | 0.4091 ($\pm$ 0.04) | **0.1550** ($\pm$ 0.02) |
| 22 clusters | 0.6331 ($\pm$ 0.03) | 0.5970 ($\pm$ 0.02) | 0.4382 ($\pm$ 0.03) | **0.1624** ($\pm$ 0.03) |

### 5.2 Organ-to-patient assignment

Given a dataset of approx. 19k patients– of which there are 14k who received a transplant organ –we test OrganITE against five other benchmark policies. We report the total population life-years

for every policy. Further statistics and results for other metrics can be found in our supplemental material.

**Benchmarks.** We formulate five benchmarks, $\pi(\mathcal{X}_Q, \mathbf{O})$: (i) best match (BM), $\pi_{\mathrm{BM}}(\mathcal{X}_Q, \mathbf{O}) := \mathrm{argmax}_{\mathbf{X} \in \mathcal{X}_Q}\{Y^{\mathbf{O}}|\mathbf{X}\}$; (ii) first in first out (FIFO), $\pi_{\mathrm{SPF}}(\mathcal{X}_Q, \mathbf{O}) := \mathrm{argmin}_{\mathbf{X} \in \mathcal{X}_Q}\{s_{\mathbf{X}}\}$; (iii) sickest person first (SPF), $\pi_{\mathrm{SPF}}(\mathcal{X}_Q, \mathbf{O}) := \mathrm{argmin}_{\mathbf{X} \in \mathcal{X}_Q}\{Y^{\emptyset}|\mathbf{X}\}$; (iv) incremental survival (IS), $\pi_{\mathrm{IS}}(\mathcal{X}_Q, \mathbf{O}) := \mathrm{argmax}_{\mathbf{X} \in \mathcal{X}_Q}\{Y^{\mathbf{O}} - Y^{\emptyset}|\mathbf{X}\}$; and (v), ConfidentMatch (CM) where the policy is the same as BM, but outcomes are predicted using [10]. A more detailed discussion on these policy benchmarks can be found in our supplemental material.

Table 3: Population life years for various policies over 10 different folds. On synthetic we use two different ITE estimators for all policies when applicable. On real data we use three different models to provide a counterfactual (CF) value (with NN as nearest neighbor to save space), as well as an ITE estimate for the policies when applicable. Standard deviation is reported in brackets below each score.

| | Used model | FIFO | SPF | BM | IS | CM | OrganITE |
|---|---|---|---|---|---|---|---|
| **Synthetic** | *ITE model* | 83509 (192.6) | 92153 (156.1) | 104889 (193.1) | 111228 (171.8) | 110129 (129.5) | **112359** (141.8) |
| | *TB* | n.a. | 71953 (132.1) | 86664 (101.4) | 81813 (125.2) | n.a. | **106392** (118.9) |
| **Real data** | *NN CF* | 94062 (157.9) | 83198 (132.1) | 100249 (164.1) | 102303 (152.1) | n.a. | **108217** (138.6) |
| | *ITE model CF* | 95646 (121.2) | 84757 (98.4) | 92948 (186.8) | 105866 (165.6) | n.a. | **107623** (154.8) |
| | *TB CF* | 89979 (172.5) | 83347 (148.3) | 99259 (122.1) | 101585 (139.6) | n.a. | **102773** (114.5) |

**Experimental setup.** For every policy, we iterate over a data-fold and present new patients and organs, as they appear. Each policy then presents a patient in the waiting queue which they deem optimal given the available organ. Whenever a patient dies– either before or after transplantation –their life-years are added to the total population life-years. Crucial is the method by which we obtain a patient's life-years: for our synthetic setup, we use the same outcome function as we used in our ITE experiments; for our real data, we use an ITE model to provide us with an outcome. A detailed algorithm depicting this setup can be found in our supplemental material.

**Results.** Table 3 demonstrates that OrganITE is the best method across various test environments. Furthermore, we can see that that prediction based policies (BM, IS, CM, OrganITE) outperform pure queue based methods. Note that results reported on synthetic data are dependent on both the policy, and the used ITE estimator; while results reported on real data, are only dependent on the policy as they use the same ITE estimator as was used to provide a counterfactual outcome when needed. As such, CM is only reported in our synthetic setup, as CM is merely a prediction method. We refer to our supplemental material for a more detailed breakdown of these results.

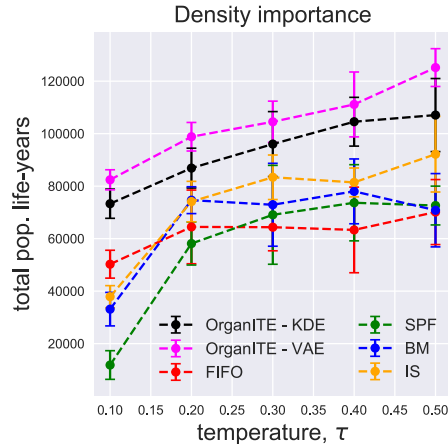

Figure 4: Population life years as a function of temperature, $\tau$, as in (7), over 10 different runs. An extensive discussion as well as a more detailed breakdown of this result— where we also report the death rates and life-years before and after transplant-surgery —can be found in our supplemental material. Error bars are 95% confidence intervals.

## 5.3 Density importance

**Experimental setup.** One of our main contributions is the explicit use of an organ-density, $p(\mathbf{O})$. In order to gain insight into how important this density is to the OrganITE policy, we investigate the effect of making some organ-types rarer on performance.

Let $\mathbf{O}_i \in c_i$ where $c_i$ is a cluster, and $i = 1, \ldots, k$, and $\mathbf{O}_i \sim \mathcal{N}_i(\boldsymbol{\mu}(c_i), \Sigma(c_i))$. We combine the Gaussian distributions in a mixture Gaussian, $\mathbf{O} \sim \mathbf{w}^T[\mathcal{N}_1, \ldots, \mathcal{N}_k]$, with $\mathbf{w} \in [0, 1]^k$ and $\sum_{i=1}^{k} \mathbf{w}_i = 1$, using the softmax distribution,

$$\sigma(\mathbf{w}) \coloneqq \frac{\exp\{p_i/\tau\}}{\sum_{c_j}^{C} \exp\{p_j/\tau\}}, \tag{7}$$

where: $\tau \in (0, 1]$ is a temperature parameter; and $p_i = \frac{|c_i|}{|\mathcal{O}|}$ is the initial probability of sampling from cluster $c_i$.

Using the temperature, $\tau$, we can alter $p(\mathbf{O})$ through $\mathbf{w}$— when $\tau$ is low, the rarity of the rarest (in the data) organ-types (clusters) is increased while the most common are made even more common. Using $\tau$ we evaluate all policies as we did in the synthetic policy experiments (in Section 5.2).

**Results.** From Figure 4, we see that OrganITE is much more resilient to more extreme conditions. While OrganITE's performance does drop given lower $\tau$, the drop is less severe when compared to other policies. We believe this is an important result as it allows OrganITE's adoption in other contexts, subject to a scarce (medical) treatment. Furthermore, we also show that other prediction based policies (SPF, BM, and IS) suffer significantly when the rarity of organs is increased, e.g., when $\tau = 0.1$.

## Broader impact

It is very clear that machine learning has the potential to transform healthcare. Its success both in other domains and already within healthcare is very promising. OrganITE has the potential to *extend lives*. However, as with almost any other machine learning method, there are risks associated with its deployment in a real healthcare setting. We would stress that to mitigate these risks, we envisage OrganITE as a *decision support tool*, with the clinicians and their patients making the final decision on any suggested organ-recipient pairing.

**A match made in high dimensions.** A persistent problem in all transplantation programs is the shortage of suitable donor organs of appropriate quality, resulting in a significant waiting list mortality. That issue is compounded by the importance of matching high-dimensional donor characteristics to high-dimensional recipient characteristics to achieve the best outcome for that donor and recipient pair. Outcome after transplantation is dependent on a wide range of factors including the cause of the disease, clinical status, and laboratory parameters of the recipient (describing how sick they are) as well as characteristics related to the donor quality including age, body mass index, donor diabetes, and other factors. The interaction of all these parameters is complex and makes the problem ripe for a machine learning driven solution [52].

**Machine learning can fail.** It is very important to note that machine learning models can fail, and OrganITE is no exception. In Section 3 we outline two assumptions that need to hold to make correct estimations of the potential outcomes used in OrganITE: overlap, and unconfoundedness. Failing to satisfy these assumptions can lead to OrganITE failing to learn the necessary individualised treatment effect, and thus assigning an organ sub-optimally with respect to its objective, resulting in the "proper" recipient potentially failing to receive an organ they should have. This has the very serious potential for mortality that would/could have otherwise been avoided. Having domain knowledge of the data is therefore crucially important [3, 53, 28] to ensure it satisfies the assumptions.

**The past is not the present.** Learning from historical data (the data used in the experiments of this paper spanned 26 years), presents further possible concerns. Surgical techniques, immunosuppressive protocols, allocation and management policies have all materially changed over the years. Biases will present differently over time. Historical data will not only reflect biases produced by medical

policy but also social policy, which has implications regarding the fairness of the learned model with respect to previously disadvantaged subgroups of the population - if access to healthcare *in the past* was difficult for certain subgroups, they may not be well represented in the data, and OrganITE may fail to learn their potential outcomes well.

**A population of equals.** Agreeing on the best objective in organ-transplantation is incredibly difficult [54]. The total life-years objective used here puts equal value in each year of each person's life. On the surface this may seem "fair", but the moral question of whether, for example, 2 people surviving for 10 additional years is better than 1 surviving for 21 additional years is deep and one that society needs to answer for itself. Does the answer change if you learn that one of the people involved is 85 while the other is 15? OrganITE, and moreover this paper, does not attempt to answer this question, nor is it tied to total life-years. OrganITE can be adapted to other objectives straightforwardly by replacing the value function associated with total life-years with another that places different weighting on different recipients' lives.

There are several different organ allocation schemes that have been considered in health systems [55–57], where the issue of the definition of equitable allocation continues to be debated. Factors such as equity concerning aetiology of organ disease [58], demographic, and deprivation status need to be considered [59], as well as geography and distance from a transplant centre [60, 61]. Whilst methodologies such as OrganITE, including its emphasis on future donor organ density, may show theoretic benefits, ensuring equity of access in those parameters will be important and will need to be considered before actual implementation.

The National Liver Offering Scheme, based on net life-years gained, transplant benefit, currently used in the UK, was introduced in March 2018 after a transplant benefit score was shown to maximise population life years in a simulation comparing transplant benefit with prior unit based allocation, a sickest first policy or a utility policy [3]. Transplant benefit policies, as compared to a simple needs-based policy also have some ethical benefits [62] as they attempt to balance both justice (prioritizing high need) and utility. Although concerns have been raised about the imprecision in calculating net benefit using Cox models, which was acknowledged by Schaubel et al. [54], our results demonstrates OrganITE's superiority compared to other policies in our simulations.

## Acknowledgements

We thank the reviewers for their helpful comments during the NeurIPS review process as they have greatly improved the paper. The research presented in this paper was supported by The Alan Turing Institute, under the EPSRC grant EP/N510129/1 and by the US Office of Naval Research (ONR), NSF 1722516.

## Footnotes

[1]When $\mathbf{X} = (x_1, x_2, \ldots, x_d)^T$ is concatenated to $\mathbf{O} = (o_1, o_2, \ldots, o_e)^T$, we create a vector, $\texttt{concat}(\mathbf{X}, \mathbf{O}) = (x_1, x_2, \ldots, x_d, o_1, o_2, \ldots, o_e)^T$.

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
