[Supplementary Material]

# A Algorithms

## A.1 Training the ITE model

In Algorithm 1 we give a detailed overview of how our ITE model (Section 4) is trained. There are two main blocks: first the organ clusters are trained; then, we train our ITE model using backpropagation. The GRL allows us to compartmentalise the gradient updates in three distinct updates: one for the outcome prediction parameters, $\theta_Y$; one for the treatment classification parameters, $\theta_c$, based on the organ clusters trained in the first block; and one for the representation parameters, $\theta_\Phi$, where the previous losses are combined as in (5). Of course, these three distinct updates are equal to one update over $\frac{\partial \mathcal{L}_{\text{ITE}}(\theta_\Phi, \theta_Y, \theta_p)}{\partial \theta_\Phi \partial \theta_Y \partial \theta_p}$, instead, but found this to abstract away to much detail at the expense of understanding. [47]

---

**Input** : model parameters, $\theta_Y, \theta_p, \theta_\Phi$
        training data, $\mathcal{D} = \{(\mathbf{x}_i, \mathbf{o}_i, y_i) : i = 1, ..., N\}$
        amount of organ types, $k$
        learning rate, $\delta$

$c(\mathbf{O}) \leftarrow \text{KMeans}(k, \{\mathbf{o}_i : i = 1, .., N\} \subset \mathcal{D})$ ;     // Training the organ-clusters
**for** $e = 1, ..., \text{max epochs}$ **do**
    **for** *Batch* $\mathcal{B} = \{(\mathbf{x}_j, \mathbf{o}_j, y_j, c(\mathbf{o}_j)) : j = 1, ..., |\mathcal{B}|\}$ *in epoch* **do**
        Compute $\mathcal{L}_{\text{MSE}}(\theta_\Phi, \theta_Y) = \frac{1}{|\mathcal{B}|} \sum_{j \in \mathcal{B}} \mathcal{L}^j_{\text{MSE}}(\theta_\Phi, \theta_Y)$ ;  // Output prediction loss
        Compute $\mathcal{L}_{\text{CE}}(\theta_\Phi, \theta_p) = \frac{1}{|\mathcal{B}|} \sum_{j \in \mathcal{B}} \mathcal{L}^j_{\text{CE}}(\theta_\Phi, \theta_p)$ ;  // Treatment classification loss
        $\theta_Y \leftarrow \theta_Y - \delta \frac{\partial \mathcal{L}_{\text{MSE}}(\theta_\Phi, \theta_Y)}{\partial \theta_Y}$ ;      // Update outcome prediction parameters
        $\theta_p \leftarrow \theta_p - \delta \frac{\partial \mathcal{L}_{\text{CE}}(\theta_\Phi, \theta_p)}{\partial \theta_p}$ ;      // Update classification parameters
        $\theta_\Phi \leftarrow \theta_\Phi - \delta \left( \frac{\partial \mathcal{L}_{\text{MSE}}(\theta_\Phi, \theta_Y)}{\partial \theta_\Phi} - \gamma \frac{\partial \mathcal{L}_{\text{CE}}(\theta_\Phi, \theta_c)}{\partial \theta_\Phi} \right)$ ;     // Update representation parameters
    **end**
**end**

**Algorithm 1:** Training our ITE model.

## A.2 Organ-to-patient assignment results

We describe how we evaluate the organ-to-patient assignment policies on real data in Algorithm 2. Should a patient-organ pair be present as-is in the test set, we base the score on the factual outcome, otherwise we use a counterfactual model to provide an outcome. Furthermore, after every organ assignment, we check whether the other patients have died while in $\mathcal{X}_Q$.

**Input** : data, $\mathcal{D} = \{(\mathbf{x}_i, \mathbf{o}_i, y_i) : i = 1, ..., N\}$
        test data, $\mathcal{D}_{\text{test}} \subset \mathcal{D}$
        counterfactual model, $\hat{Y}$
        to-evaluate policy, $\pi$
**for** $t = 0, 1, ..., |\mathcal{D}_{test}|$ **do**
    $\mathcal{X}_Q^t \leftarrow \mathcal{X}_Q^{t-1} \cup \{\mathbf{X}^t\} \in \mathcal{D}_{\text{test}}$;
    $\mathbf{o} \leftarrow \mathbf{O}^t \in \mathcal{D}_{\text{test}}$;
    **if** $\mathbf{o} \neq \emptyset$ **then**
        $\mathbf{x} \leftarrow \pi(\mathbf{o})$;
        **if** $(\mathbf{x}, \mathbf{o}) \in \mathcal{D}_{test}$ **then**
            score $\leftarrow$ score $+ \mathcal{D}_{\text{test}}(Y|\mathbf{X}, \mathbf{O})$ ;       `// Patient-organ is in test-set`
        **else**
            score $\leftarrow$ score $+ \hat{Y}(\mathbf{X}, \mathbf{O})$ ;     `// Patient-organ pair not in test-set`
        **end**
    **end**
    **for** *all* $\mathbf{x} \in \mathcal{X}_Q^t$ **do**
        **if** $\mathbf{x}$ *died in* $\mathcal{X}_Q^t$ **then**
            **if** $\mathbf{x}$ *did not receive an organ in* $\mathcal{D}_{test}$ **then**
                score $\leftarrow$ score $+ \mathcal{D}_{\text{test}}(Y|\mathbf{X})$ ;     `// Patient-organ is in test-set`
            **else**
                score $\leftarrow$ score $+ \hat{Y}(\mathbf{X})$ ;     `// Patient-organ pair not in test-set`
            **end**
        **end**
    **end**
**end**

**Algorithm 2:** Evaluation of a policy, $\pi$. $\hat{Y}$ is trained using $\mathcal{D}_{\text{train}}$, where $\mathcal{D}_{\text{test}} \cup \mathcal{D}_{\text{train}} = \mathcal{D}$ and $\mathcal{D}_{\text{test}} \cap \mathcal{D}_{\text{train}} = \emptyset$.

## B   Benchmarks and ablations: details

### B.1   ITE model

We compare our ITE model with ConfidentMatch, and a multitask network which predicts the outcome per organ-type (based on the organ clusters, $c(\mathbf{O})$).

**ConfidentMatch [10, 28].** ConfidentMatch is an ensemble prediction method where the trainingdata is divided in partitions and a predictor method is fitted on every partition. Given an hypothesis space (i.e., prediction methods of a certain VC-dimension), and a maximum partition count, ConfidentMatch optimises a prediction loss over different compositions of the partitions and combinations of predictors from the hypothesis space.

We set the maximum partition count to the number of organ-clusters we used when comparing to our ITE model. For example, when we fit our KMeans cluster with $k = 15$, we restrict ConfidentMatch's partition count to 15. Furthermore, we set the hypothesis space to a `RandomForestRegressor`, support vector machine for regression (SVR), and a multi-layered perceptrion (MLP) [63].

**Multi-task network.** Given the organ-clusters, $c(\mathbf{O})$, we train a multi-task network to predict outcomes for every organ-cluster. Specifically, we employ a hard-parameter sharing methodology for our mutli-task network. As we have only one factual outcome for every patient, we set the counterfactuals (i.e. the other organ-clusters) to the prediction when computing the loss. As such,

(a) Padded network.      (b) Siamese network.      (c) Two networks.

Figure 5: Networks used in the ablation study.

the loss for the counterfactuals remains $0$, such that the gradient update is only based on the factual outcome.

## B.2 Ablation study networks

As we have in Figure 3, we illustrate the architectures for the ablation studies of our ITE model in Figure 5.

## B.3 Organ-to-patient assignment policy benchmarks

**Best match (BM)** $-\pi_{\text{BM}}(\mathcal{X}_Q, \mathbf{O}) \coloneqq \operatorname{argmax}_{\mathbf{X} \in \mathcal{X}_Q} \{Y^{\mathbf{O}} | \mathbf{X}\}$

A common organ-to-patient assignment policy, selecting whomever is associated with highest life expectancy, given the available organ. We use the same policy for ConfidentMatch, but let $\hat{Y}$ be estimated as in Yoon et al. [10].

**First in first out (FIFO)** $-\pi_{\text{FIFO}}(\mathcal{X}_Q, \mathbf{O}) \coloneqq \operatorname{argmin}_{\mathbf{X} \in \mathcal{X}_Q} \{s_{\mathbf{X}}\}$

FIFO is a naive scheduling algorithm that simply selects the oldest addition to the queue, based on $s_{\mathbf{X}}$ representing the time of entry, whenever an organ becomes available.

**Sickest person first (SPF)** $-\pi_{\text{SPF}}(\mathcal{X}_Q, \mathbf{O}) \coloneqq \operatorname{argmin}_{\mathbf{X} \in \mathcal{X}_Q} \{Y^{\emptyset} | \mathbf{X}\}$

Like FIFO, we relate SPF to common queueing strategies, where SPF relates to a prioritised queue. While a measure of sickness is not directly observable, we can approximate it with a patient's estimated life expectancy, $\hat{Y}^{\emptyset}$. That is, a patient with lower life expectancy is considered sicker than a patient with higher life expectancy.

**Incremental survival (IS)** $-\pi_{\text{IS}}(\mathcal{X}_Q, \mathbf{O}) \coloneqq \operatorname{argmax}_{\mathbf{X} \in \mathcal{X}_Q} \{Y^{\mathbf{O}} - Y^{\emptyset} | \mathbf{X}\}$

Currently employed as policy in the UK, is an estimate of incremental survival rates for an individual patient [9]. In effect, this is a first step towards a counterfactual based approach, though it should be noted that in Neuberger et al. [9], assignment bias was not balanced from the dataset.

# C   Additional results

## C.1   Organ-to-patient assignment

We present a detailed breakdown of the results for our organ-to-patient assignment policy. Specifically, we report: (i) the premature deaths in $\mathcal{X}_Q$ and $\mathcal{X}_M$— where any death in $\mathcal{X}_Q$ and any death before 5 years in $\mathcal{X}_M$ is considered premature; and (ii) the average time alive in $\mathcal{X}_Q$ and $\mathcal{X}_M$— where patients in $\mathcal{X}_M$ are not considered for the average time alive in $\mathcal{X}_Q$.

While these results are informative on the performance of all policies, they require careful attention. For example, a high life expectancy in $\mathcal{X}_Q$ is not necessarily a good property of a policy, as this means that potentially healthier patients are dying before they receive a transplant-organ. Similarly for deathrates in $\mathcal{X}_Q$, where a low deathrate in $\mathcal{X}_Q$ could indicate a waste of tranplant-organs as the policy might prefer sicker patients, with lower life expectancy in $\mathcal{X}_M$.

As mentioned in our related works (Section 2), deciding the objective brings forth an ethical discussion and should be done with great care. By reporting these details, we offer argumentation for clinicians interested in implementing a policy for scarce medical resources. While OrganITE is clearly the best policy when optimising the total population life years (as was our objective, cfr. Section 3), some situations might favor SPF as it allows sicker patients to be treated first. An example of such a situation could be pain relief, where patients suffering the most are aided first.

Table 4: Organ-to-patient evaluation on synthetic data over 10 different folds. Lower is better above the dotted line, and higher is better below the dotted line.

| *Using our ITE model* | FIFO | SPF | BM | IS | CM | OrganITE |
|---|---|---|---|---|---|---|
| Population life years | 83509 | 92153 | 104889 | 111228 | 110129 | **112359** |
| Deaths in $\mathcal{X}_Q$ | 0.2646 | 0.2309 | 0.2357 | 0.2067 | 0.2038 | **0.1926** |
| Deaths before 5 years in $\mathcal{X}_M$ | 0.1683 | 0.1869 | 0.1702 | 0.1593 | 0.1891 | **0.1472** |
| Avg. days alive in $\mathcal{X}_Q$ | 32.49 | 32.38 | 32.81 | 32.65 | 33.12 | **37.19** |
| Avg. years alive in $\mathcal{X}_M$ | 4.347 | 4.138 | 5.088 | 5.057 | 5.165 | **5.905** |
| *Using TransplantBenefit* | | | | | | |
| Population life years | n.a. | 71953 | 86664 | 81813 | n.a. | **106392** |
| Deaths in $\mathcal{X}_Q$ | n.a. | **0.1587** | 0.2179 | 0.3201 | n.a. | 0.3346 |
| Deaths before 5 years in $\mathcal{X}_M$ | n.a. | 0.3201 | 0.3152 | **0.3048** | n.a. | 0.3055 |
| Avg. days alive in $\mathcal{X}_Q$ | n.a. | 24.46 | 11.55 | 20.52 | n.a. | **31.50** |
| Avg. years alive in $\mathcal{X}_M$ | n.a. | 4.222 | 5.181 | 4.572 | n.a. | **5.785** |

**Results on synthetic data.** In Table 4 we report a detailed breakdown of the results presented in the upper part of Table 3 in our main text. From this we learn that OrganITE's performance is better across all reported metrics when using our ITE model. However, when using TransplantBenefit we notice weaker performance in death rates, especially in $\mathcal{X}_Q$. This small performance drop is made up for by a significant increase in expected life years after transplantation.

**Results on real data.** We argue that OrganITE's performance is a result of balancing the various aspects taken into account in organ-to-patient assignment (cfr. Section 4.1). Leveraging this balance results in less deaths and high life expectancy, making OrganITE such a successful assignment-policy.

This balance is visible in Table 5, where we find OrganITE to excel in life expectancy post-transplantation, while maintaining decent performance in the other performance indicators. For example, notice how OrganITE is best in life expectancy post-transplantion across all counterfactual models, while performing: best or second best in premature deaths in $\mathcal{X}_M$; never worst in death rates for $\mathcal{X}_Q$ (even second best for TB as the counterfactual model); and never worst in life expectancy in $\mathcal{X}_Q$ (third using TB as the counterfactual model).

Furthermore, notice how SPF has higher life expectancy and lower death rates in $\mathcal{X}_Q$, while performing very poorly in total population life years. SPF's performance is due to the aforementioned greedy approach to selecting the sickest patients in the waiting queue, $\mathcal{X}_Q$.

Table 5: Organ-to-patient evaluation on real data over 10 different folds. Lower is better above the dotted line, and higher is better below the dotted line.

| *Nearest neighbor counterfactual* | FIFO | SPF | BM | IS | OrganITE |
|---|---|---|---|---|---|
| Population life years | 94062 | 83198 | 100249 | 102303 | **108217** |
| Deaths in $\mathcal{X}_Q$ | 0.4414 | **0.4055** | 0.4209 | 0.4236 | 0.4233 |
| Deaths before 5 years in $\mathcal{X}_M$ | 0.2196 | 0.2422 | 0.2093 | 0.1972 | **0.1787** |
| Avg. days alive in $\mathcal{X}_Q$ | 28.78 | **28.96** | 27.83 | 28.12 | 28.15 |
| Avg. years alive in $\mathcal{X}_M$ | 3.805 | 3.363 | 4.057 | 4.138 | **4.378** |
| *ITE model counterfactual* | | | | | |
| Population life years | 95646 | 84757 | 92948 | 105866 | **107623** |
| Deaths in $\mathcal{X}_Q$ | 0.4294 | **0.4081** | 0.4189 | 0.4245 | 0.4257 |
| Deaths before 5 years in $\mathcal{X}_M$ | 0.1995 | 0.2455 | 0.1996 | **0.1794** | 0.1806 |
| Avg. days alive in $\mathcal{X}_Q$ | 28.70 | **28.96** | 28.33 | 28.13 | 28.05 |
| Avg. years alive in $\mathcal{X}_M$ | 3.875 | 3.436 | 3.765 | 4.282 | **4.349** |
| *TB counterfactual* | | | | | |
| Population life years | 89979 | 83347 | 99259 | 101585 | **102773** |
| Deaths in $\mathcal{X}_Q$ | 0.4294 | **0.4041** | 0.4277 | 0.4167 | 0.4113 |
| Deaths before 5 years in $\mathcal{X}_M$ | 0.2195 | 0.2653 | 0.2296 | 0.1994 | **0.1906** |
| Avg. days alive in $\mathcal{X}_Q$ | 27.21 | **28.83** | 22.34 | 25.48 | 26.77 |
| Avg. years alive in $\mathcal{X}_M$ | 3.676 | 3.348 | 4.008 | 4.109 | **4.153** |

Figure 6: Policy performance indicators in function of temperature, $\tau$, as in (7). From left to right: Deaths in $\mathcal{X}_Q$, Deaths in $\mathcal{X}_M$, Days alive in $\mathcal{X}_Q$, and Years alive in $\mathcal{X}_M$. For deaths, lower is better; for time alive, higher is better.

## C.2 Density importance

Adjusting the density $p(\mathbf{O})$ in the synthetic experiment (Table 4) allows us to report on how important this density is to OrganITE and other organ-assignment policies, in terms of the performance metrics presented above. From Figure 4 we learn that OrganITE's expected total population life years seems much less affected by more extreme organ-densities, when compared to the other organ-to-patient assignment policies. Here, we report the same breakdown of our result as we have above, providing a more detailed argument for the organ-to-patient assignment policies.

**What is a more extreme density?** First, we clarify that by introducing a temperature parameter, we make some organs more rare than other organs. Specifically, because the weights, $\mathbf{w}$, in (7) sum to one ($\sum \mathbf{w} = 1$ making $0 \leq \mathbf{w}_i \leq 1 \forall \mathbf{w}_i$), and every component of $\mathbf{w}$ is divided by $\tau \in (0, 1]$, we make some organ clusters less likely or more likely. As such, leaving patients in need of otherwise more frequent organs (as they are present in $\mathcal{D}$), now less likely to receive a suitable organ.

**Why is this important?** Having more patients in need of rare organs, requires a policy to explicitly handle these deficits in a way that gives higher priority to these patients, at the cost of potentially multiple better matches with the available organ. From this experiment it is clear that using an estimate of $p(\mathbf{O})$ allows OrganITE to make more informed decisions on how to distribute organs in such an extreme condition.

**Results.** Consider Figure 6 where we breakdown the result presented in Figure 4 as we did for the results above. From this breakdown we notice that OrganITE's performance does drop when the conditions get more severe, however, they remain much more stable when compared to the other policies.

Another observation we make is how FIFO seems largely unaffected in $\mathcal{X}_Q$. We argue this is due to FIFO being a queue based policy, effectively assigning organs to patients without any estimate of $\hat{Y}$. Naturally, FIFO is triumphed in performance by all estimation based policies, i.e., IS, BM, SPF, and OrganITE; though this might explain why FIFO is performing slightly better in Figure 4 given low $\tau$. Furthermore we notice that IS' performance is fairly close to OrganITE's performance when $\tau$ increases, suggesting the density might matter less when the organ-density converges to a uniform density. We reason as such, as IS resembles OrganITE most closely as it too uses an ITE estimation rather than mere prediction.

# D   Hyperparameters

Table 6: Hyperparameters ITE model and ablations

|  | Padded network | Two models | Siamese network |
|---|---|---|---|
| *Layers* | • $\Phi$:(32, 16, 16) <br> • $c$:(16, 16, $k$) <br> • $Y$:(16, 16, 1) | • $\Phi^{\emptyset}$:(16, 16, 16) <br> • $\Phi^{\mathbf{O}}$:(32, 16, 16) <br> • $c$:(16, 16, $k$) <br> • $Y^{\emptyset}$:(16, 16, 1) <br> • $Y^{\mathbf{O}}$:(16, 16, 1) | • $\Phi$:(16 and 32, 16, 16) <br> • $c$:(16, 16, $k$) <br> • $Y^{\emptyset}$:(16, 16, 1) |
| *Activation* | ReLU | ReLU | ReLU |
| *Learning rate* | 0.0004 | $\emptyset$: 0.0001; $\mathbf{O}$:0.0004 | 0.0004 |
| $\gamma$ | 0.25 | 0.25 | 0.25 |
| *max-epochs* | 60 | 60 | 60 |
| *batch-size* | 100 | 100 | 100 |

Table 7: Hyperparameters multitask network

| Layers | activation | learning rate | max-epochs | batch-size |
|---|---|---|---|---|
| (32, 16, 16, 16, 16, $k$) | ReLU | 0.0004 | 60 | 100 |

Table 8: Hyperparameters OrganITE

| $a$ | $b$ | $\alpha_1$ | $\alpha_2$ | KDE bandwith | KDE kernel |
|---|---|---|---|---|---|
| 1 | 1 | 1 | 1 | 1 | Gaussian |

# E Data

Table 9: Features used in real data experiments.

| Recipient | | Organ | | Cause of death | |
|---|---|---|---|---|---|
| Name | Mean (std.) | Name | Mean (std.) | Cause | Proportion |
| Height | 168.6 (17.6) | BMI | 25.8 (4.95) | Intracranial haemorrhage | 57.4% |
| Gender | 37.3% male | Gender | 53.3% male | Hypoxic brain damage - all causes | 13.8% |
| Haemoglobin | 11.5 (3.8) | Cause of death | see right | Other trauma - accident | 3.4% |
| White blood cells | 5.6 (3.8) | Age | 46.8 (15.99) | Intracranial - type unclassified (CVA) | 3.3% |
| Platelets | 123.5 (90.5) | Donor type | 84.7% brain dead | Unspecified | 3.1% |
| Serum urea | 6.3 (5.6) | | 13.7% circulatory death | Trauma RTA - car | 2.9% |
| Serum creatinine | 84.9 (43.7) | | 1.13% living | Intracranial thrombosis | 2.1% |
| Serum albumin | 31.9 (6.7) | | 0.39% domino | Trauma RTA - pedestrian | 1.8% |
| INR | 1.4 (0.5) | | | Living donor | 1.4% |
| Serum bilirubin | 87.0 (119.0) | Meningitis | 1.4 % | | |
| Serum sodium | 136.2 (4.8) | Brain tumour | 1.4% | | |
| Serum potassium | 4.2 (0.53) | Trauma RTA - motorbike | 1.1% | | |
| PO2 | 12.5 (3.44) | | | Under 1% not reported. | |
| AFP level | 26.0 (286.37) | | | | |