[Reviews · NeurIPS 2020]

Review 1

Summary and Contributions: This paper considers an organ allocation setting, where a fixed set of patients (organ recipients) are matched by a policy with organs, which arrive one-by-one. The authors propose a policy that aims to maximize total life years, by considering both quality of the match between the patient and organ, and the rarity of the organ.The paper is clearly written, and well-cited. Notation is consistent and fairly easy to follow. The use of ITE in this setting appears to be quite novel and difficult; it seems that the entire paper could have been focused on ITE (in fact, I would likely have given a more positive review in this case, see below). The authos clearly present their method and discuss their implementation and hyperparameters; their experiments clearly demonstrate an improvement over similar methods.

Strengths: The paper is clearly written, and well-cited. Notation is consistent and fairly easy to follow. The use of ITE in this setting appears to be quite novel and difficult; it seems that the entire paper could have been focused on ITE (in fact, I would likely have given a more positive review in this case, see below). The authors clearly present their method and discuss their implementation and hyperparameters; their experiments clearly demonstrate an improvement over similar methods.

Weaknesses: The authors first state that they set aside the moral dilemmas of allocating scarce organs, and instead focus only on total expected life years. I believe this is a perfectly good approach. However, in sec. 4.1 the authors begin discussing moral implications of their policy - which may be impacted by the choice of hyperparameters. (Furthermore, the various supervised and unsupervised methods used throughout the paper likely have different impacts on different sets of patients, e.g., patients who aren't well-represented in the training data.) All of the above issues likely warrant further discussion in the paper, due to the inherently ethical nature of organ allocation. At very least, I do not believe the authors can excuse themselves from issues of morality (2nd paragraph of the paper: "The discussion of which objective is morally "correct" is well outside the scope of this paper..."), since they choose an objective with inherent moral implications in sec. 4.1. Rather, I suggest the authors acknowledge that these issues are complicated, and this work is no exception. The authors do not provide a dataset or code to implement their methods. If this code and data is available, I could not find this in the paper or supplement. (Of course I don't expect sensitive medical data to be made public, but some experiments involve synthetic data, and perhaps this data generator can be shared.) Finally, the statement that their method, "OrganITE, [is] the first organ-assignment-policy that explicitly takes into account the rarity of an organ" is simply not true. Matching-based approaches naturally account for the scarcity of certain types of organs--e.g., see the distinction between over- and under-demanded organs used in several papers, such as (Price of Fairness in Kidney Exchange, Dickerson et al. 2014). Furthermore, existing organ allocation policies (such as OPTN policies used in the US) prioritize transplants to highly-sensitized patients, who are less likely to find a compatable organ; these policies aim to ensure that when a "rare" organ arrives (which is compatible with a highly-sensitized recipient), it is allocated to a patient who can receive only this type of rare organ. On reproducibility: The authors describe the algorithms used to implement their methods. I did not see any mention of a codebase or dataset that the authors would provide to reproduce their work. Since data is necessary to their approach, I would not be able to reproduce their methods & experiments.

Correctness: The only claim in this paper that seems incorrect is that they are the first to incorporate organ rarity into their allocation policy (see above). The experimental results are very compelling; the results are plausible, and none of the mathematical claims appear incorrect. However I am not an expert in ITE.

Clarity: Yes, this paper is mostly very clear, but I fear it would be difficult to follow for someone not familiar with organ allocation and machine learning.

Relation to Prior Work: Yes. The authors discuss several related work in the ML space, but very little discussion of non-ML organ allocation literature.

Reproducibility: No

Additional Feedback:


Review 2

Summary and Contributions: The paper presents a new framework for optimizing organ allocation that involves estimating ITE of giving an organ to a particular individual as well as considering the scarcity of the organ and possibility of better organs existing for the individual in question. Performance improvements in terms of expected life-years are shown on synthetic and real data. I think the rebuttal did a good job addressing many of my concerns; even in cases where a perfect answer is difficult (ie, how to evaluate life-years saved under a new policy) I think the methodology and responses to criticism are well thought-out. I do ask, as some other reviewers did, that the authors spend some space addressing the moral and ethical complexities of this topic in the final version. My score remains the same.

Strengths: The combination of new ML approaches for estimating ITE with organ donation appears entirely novel, as does the more sophisticated objective and attempt to maximize total life-years. Empirical improvements are demonstrated both in ITE estimation and in overall life-year gains. Overall, the framework seems like a valuable contribution in this area, as well as the approach to a high-dimensional treatment.

Weaknesses: ---Main concerns: --- It appears a lot of effort was put into evaluation, but I still feel there are some limitations that may be inevitable re: estimating ITEs. In particular, the evaluation on real data, in terms of life-years, itself relies on an estimator of ITE. Further details here would be useful, as if this is the same estimator used in the model it could inflate performance estimates. The link from the given end-objective (life-years) to the particular optimization problem in (1) appears heuristic; intuitively I see why optimizing each term of (1) is good but is anything proven about the relationship between this objective function and the stated goal of life-years? Given that evaluation involves simulating the process of organ allocation and calculating the expected life years resulting from a policy, why not use a reinforcement learning approach to directly optimize for this outcome? ---More minor concerns: --- Why use a standard autoencoder + KDE to estimate density? If this is a standard approach, I am not aware; I am more familiar with VAE-type methods being used to learn a distribution. Also, my understanding is that under many rules of thumb (ie, the Silverman estimator), optimal bandwidth sharply falls off (ie, to the fifth power) as data size grows, which would affect the parameter choices. Error bars or p-values on all results (which should be possible given multiple folds) would make the experimental results much more impresive.

Correctness: Experiments seem well-designed, with ablation on both the overall approach to estimating ITE, ablation on the network architecture, and comparisons to other methods for organ allocation. Some concerns with methodology and evaluation are noted in "weaknesses".

Clarity: The paper is very clear and well-written.

Relation to Prior Work: I found the discussion of related work extensive and accurate.

Reproducibility: Yes

Additional Feedback: The impact section did not affect the numeric score; however, because organ allocation decides who should get a scarce, life-saving resource, I had hoped for more discussion of possible problems with this approach. I think there's some good discussion of this in the Supplement that didn't make it into the maintext. In addition to the Supplement's observation that there may be clinically useful objectives other than expected life years, a model that underpredicts ITE in minority groups could lead to problematically reduced gains in life expectancy among those groups. Interpretability tools are often used to understand whether such issues arise, but interpretability could be more challenging with this model than a simpler ITE estimator, etc. There are potentially other, similar concerns! None argue against the work's value, but I think the impact section is a new opportunity for discussion of such concerns.


Review 3

Summary and Contributions: The authors present a method for assigning organs to patients in a transplant queue, designed to optimize for mean life expectancy of the patient population. In contrast to generally more naive approaches used in current practice (e.g. first-come first-serve, sickest-first, or best patient match), the proposed approach balances the time that a patient may have left and the chance that another suitable organ may arrive in time. The authors show improvement (of mean population life expectancy) over several policies using both synthetic and real data, and further show superiority of their method under more extreme circumstances of organ availability. The paper is reasonably clear, though is notably lacking in-depth description of the datasets used/generated here. Nevertheless, the method is promising and seems sound, and the potential impact is high. I recommend acceptance.

Strengths: The methods and evaluation are sound and the potential biomedical impact is quite high. Relevance to the NeurIPS community at large is somewhat limited as this is certainly not a general method, but this will certainly be relevant to members of the NeurIPS community interested in applications of ML to the biomedical domain. Given the application of organ transplants, I think it could also be of interest to those focused on fairness in ML more generally.

Weaknesses: There is a lack of in-depth discussion of the datasets themselves that I find less satisfying, though I imagine this is in part driven by space limitations. I would still appreciate more detailed description of the datasets in the supplement, including patient and organ features considered. This paper honestly might be more amenable to longer form journal where further details regarding feature importances and the like could be discussed.

Correctness: Organ transplants are outside my realm of expertise, but I generally find the methodology correct. I was at first a bit dubious about the use of a counterfactual model as part of the scoring mechanism on real data (rather than simply evaluating on samples where the model agreed with the data), but the authors did at least offer multiple counterfactual models, at least one of which corresponds with a model in current practical use. I'm satisfied with that.

Clarity: The paper was clearly written.

Relation to Prior Work: Organ transplant assignment is outside my expertise, so I may be missing background, but the prior work seems sufficiently covered.

Reproducibility: Yes

Additional Feedback: -Post Response- Reviewer 1 makes some good points, and more discussion of ethical and practical concerns would improve the paper. I still lean toward acceptance. ---------------------- As I said, I would like more description of both the real and synthetic data. I would also like to see more justification of using a counterfactual model when scoring on real data, rather than simply comparing when the model and data agreed (a la https://www.ncbi.nlm.nih.gov/pmc/articles/PMC1914366/). Regardless, good work.


Review 4

Summary and Contributions: Authors propose using the Neyman-Rubin potential outcomes framework to the organ transplant assignment problem. It appears they use the framework due to the fact a patient can only receive a single organ at a given time. The authors propose a neural network-like architecture that uses backpropagation as the estimator and demonstrate improved performance over other algorithms like FIFO, SPF, and BM.

Strengths: Authors recognize the organ assignment problem as one that is inherently a missing data problem. Furthermore, it is missing in that at any given time, we can only observe one outcome (and thus uses the potential outcomes framwork). This proposal is a novel approach and application of the Neyman-Rubin POs framework.

Weaknesses: The authors need to devote a lot more time to further expand on Assumptions 1 and 2 (normally referred to as Strong Ignorability (Rubin 1984) ). For instance, Assumption 2 states that conditional on a set of covariates X, the treatment (organ) is independent of the potential outcomes. In other words, if we were to compare patients given the set of X, it can essentially be thought of as an randomized experiment. However, the authors make no mention of what X is, and it appears X is only referred to as the "patients". For example, do the authors believe Assumptions 1 and 2 hold on income, age, etc? These are strong assumptions that are normally not empirically verifiable from the data and needs much more discussion. It is also not clear why it is necessary to invoke the potential outcomes framework. Normally, usage of the potential outcomes framework should include a discussion of sensitivity analysis regarding Assumptions 1 and 2 (you can refer to Paul Rosenbaum's work here)). Furthermore, more needs to be discussed about the matching procedure used. For example, was it by replacement, without, etc? Generally, the potential outcomes framework is mentioned but requisite details are missing.

Correctness: Author mentions they use the "Neyman-Rubin" potential outcomes framework, but in line 166, they refer to the potential outcome (which appears to be the conditional mean) using do-calculus, which was originated by Judea Pearl. Author doesn't reference Pearl's literature and needs to clairfy their framework.

Clarity: The paper makes multiple mentions to neural networks (activations, layers, auto-encoders) but lacks any description of the type of architectures used. The authors should clarify the technical details of their neural network models and its relation to the aforementioned problem. For instance, in Figure 3, the authors claim "we concatenate the organ, O, to the patient, X, to learn a representation, φ." This needs to be elaborated on in detail as to how this concatenation is performed. They also mention "padded", "siamese", and "two models" without sufficient detail as to what the layer details refer to (ie. (32, 16, 16)?)

Relation to Prior Work: Authors sufficiently mention previous work in this area (both theoretically and application).

Reproducibility: No

Additional Feedback: Missing major details on the dataset used and model architecture descriptions. UPDATE: The authors sufficiently addressed my concerns regarding details in the paper and I've revised the score to a 6.

[Author Response · NeurIPS 2020]

*We thank all the reviewers for their thoughtful comments, they are much appreciated. We will first address some general comments and then respond to comments made by the individual reviewers.*

**[General].** **Data and code.** ([R1, R3, R4]) To facilitate reproducibility of our results we will release all our code (incl. synthetic data generation) upon acceptance. As the data includes confidential medical information, we can only share our data upon request. However, we will append a table with some descriptive statistics of our data (amount of patients/organs, mean and std of patient and organ covariates, disease cohort sizes, etc.), as well as an overview of the features used for patients and organs. **Evaluation using ITE.** ([R2, R3]) Should OrganITE be the only policy sharing an ITE model with evaluation, we agree with the reviewers that OrganITE would be at an unfair advantage. However, this is not the case: every (estimation-based) policy shares the same ITE estimator, this way the policies are evaluated solely based on the policy, not on how well survival is estimated (see lines 283–286). **Morality and ethics.** ([R1, R2]) We agree that this work has significant moral implications, though we want to emphasize that such implications also hold for other algorithms and even clinicians themselves. However, as was indicated by [R2], we will move some of our discussion from the supplemental material to the main text, as we are allowed a 9$^{th}$ page for a camera-ready version. Furthermore, as was suggested by [R1], we will acknowledge (in the introduction) that our method is no exception to ethical concerns.

**[R1].** **Organ assignment policy taking into account rarity of organ.** We agree with the reviewer that our statement in the conclusion about our method being the first to take into account the rarity of an organ may have been too general. What we meant to emphasize is that our organ-assignment policy learns in a data driven way what the optimal organ is for each patient, i.e the organ yielding the highest ITE for the patient (by considering the high dimensionality of the organ features) as well as the rarity of this optimal organ, by calculating its probability using an estimated density of all organs. This probability is estimated from the data. Note that this approach is different from the matching based method proposed by Dickerson et al. (2014) which considers a rule-based approach for defining organ compatibility in terms of blood groups which subsequently determines the rarity of certain organs (i.e. $\mathbb{I}_{\{v_s(\mathbf{X})>\tau\}}$, where $v_s, \tau \in [0, 100]$ are a patient's sensitization level and threshold respectively, while we define rarity as a distribution, $p(\mathbf{O}_\mathbf{X}^*)$). Similarly, the OPTN policies used in the US also use a rule-based approach to define the organ rarity in terms of its compatibility. We will revise our claim in the paper, include a discussion and cite the relevant works pointed out by the reviewer.

**[R2].** **Density estimation.** We agree that using a VAE is more standard in literature. As such, we have rerun our experiments using a VAE and will include these results in the main text. From Figure 1 we see that OrganITE with a VAE compares favorably to OrganITE using a KDE. **Optimisation target and reinforcement learning (RL).** We acknowledge that the optimisation target is heuristic. While one (with some effort) could indeed use RL to optimise for total life-years directly, we are convinced our method offers some advantages over RL: (i) explicitly taking into account an ITE estimator (and other components) allows us to interpret decisions made by OrganITE, as well as ease debugging of peculiar suggestions; (ii) for RL we would consider choosing a patient as the action, and the organ as the state resulting in a very sparse action space (as the actions are constrained to the patients currently in $\mathcal{X}_Q^t$) with minimal control over state-transitions, all resulting in an extremely hard to learn policy from logged data. However, we wish to stress that our arguments do not make it *impossible* to use RL in this setting, though it would result in a different paper entirely. **Error bars.** Not including confidence intervals (CIs) in the tables was an oversight for which we apologize. We will augment all our results with a 95% CI, as we have in our final experiment (Figure 4).

**[R3].** We believe all concerns are responded to in the [General] section above.

**[R4].** **Notation.** On line 121: $\mathcal{X} \subset \mathbb{R}^d$ is the *set* of possible patients (a subset of the real-vector space); on line 122: $\mathbf{X} \in \mathcal{X}$ is a random variable representing *one* patient as a $d$-dimensional real vector (with $\mathbf{x}$ a realisation of $\mathbf{X}$). Using this notation, Assum. 2 means that *all patient covariates (not set of patients)* $\mathbf{X}$ *that affect the treatment assignment,* $\mathbf{O}_\mathbf{X}^{\pi_{obs}}$, *and potential outcomes,* $Y^\mathbf{o}$, *are observed.* **Validity of assumptions.** While our assumptions are standard practice in ITE literature, we agree with [R4] that some assurance on whether or not these assumptions hold is warranted as they are *only* verifiable through domain knowledge. As such, we will note that: organ transplantation is a highly monitored setting where the many variables in our data are decided upon by (highly trained) clinicians. Note that we will append a description of our data as was mentioned above. **Potential outcomes (POs).** We will cite Pearl [1] as on p.166 he shows POs are shown to be equivalent to his framework, allowing the use of $do(\cdot)$. **Matching.** Our policy matches organs to patients without replacement; see lines 128–133. **Architectural details.** (32, 16, 16) indicates a 3-layered, fully-connected network, having widths 32, 16, and 16, respectively, which we will note in the supplemental material. In Figure 3 we concatinate $\mathbf{x} = (x_1, ..., x_d)^T$ and $\mathbf{o} = (o_1, ..., o_e)^T$ as $(x_1, ..., x_d, o_1, ..., o_e)^T$, which we will clarify.

Figure 1: Density importance, with OrganITE - VAE

**Reference:** [1] Judea Pearl. Causal inference in statistics: An overview. *Statistics surveys*, 3:96–146, 2009.

[Meta-Review · NeurIPS 2020]

This work considers estimating individual treatment effect (ITE) of organ donations --- with respect to scarcity and quality of match for an organ --- in order to maximize mean life expectancy of a population. Common approaches typically include first-come first-serve, local best patient matching, or most acute. Reviewers were uniformly interested in this application area, and the relative potential of machine learning to improve our collective ability to facilitate organ donations. Additionally, the paper overall is well written and reasoned. I would greatly encourage the authors to take into account feedback from the reviewers. In particular, given grounding in this particular application it is important to elaborate on the potential problems of this approach as well as the likelihood that this will be able to impact existing approaches to assignment. For the latter question, it's likely that in many geographies even an oracle would be dismissed for the perceived equity of existing strategies. This is not a fault of the method, but provides context to readers about the potential impact. Further, toward the point about deferring scope of morality: it is indeed better to acknowledge the complicated nature of this topic, and state assumptions objectively so that the reader can form the appropriate opinion. This is explicitly mentioned in the author response, but it is worth highlighting given the attention given to moral and ethical implications of submissions. NOTE FROM PROGRAM CHAIRS: This paper is given a CONDITIONAL ACCEPT. Using a purely ML-based method to allocate organs raises some clear and substantial ethical concerns. For the camera-ready version, please expand your broader impact statement to include a more thorough discussion of the potential risk of harm, including risk of failure of the method, limitations of learning from outdated historical data, etc. This is a complex question, but it must be addressed for the paper to be suitable for publication. ******************************* Note from Program Chairs: The camera-ready version of this paper has been reviewed with regard to the conditions listed above, and this paper is now fully accepted for publication.